# Novel Drug Delivery Systems as an Emerging Platform for Stomach Cancer Therapy

**DOI:** 10.3390/pharmaceutics14081576

**Published:** 2022-07-29

**Authors:** Umme Hani, Riyaz Ali M. Osmani, Sabina Yasmin, B. H. Jaswanth Gowda, Hissana Ather, Mohammad Yousuf Ansari, Ayesha Siddiqua, Mohammed Ghazwani, Adel Al Fatease, Ali H. Alamri, Mohamed Rahamathulla, M. Yasmin Begum, Shadma Wahab

**Affiliations:** 1Department of Pharmaceutics, College of Pharmacy, King Khalid University, Abha 62529, Saudi Arabia; myghazwani@kku.edu.sa (M.G.); afatease@kku.edu.sa (A.A.F.); aamri@kku.edu.sa (A.H.A.); rahapharm@gmail.com (M.R.); yaminimp47@gmail.com (M.Y.B.); 2Department of Pharmaceutics, JSS College of Pharmacy, JSS Academy of Higher Education and Research (JSS AHER), Mysuru 570015, Karnataka, India; riyazosmani@gmail.com; 3Department of Pharmaceutical Chemistry, College of Pharmacy, King Khalid University (KKU), Abha 62529, Saudi Arabia; sahussain@kku.edu.sa (S.Y.); hissanachem@yahoo.com (H.A.); 4Department of Pharmaceutics, Yenepoya Pharmacy College and Research Centre, Yenepoya (Deemed to Be University), Mangalore 575018, Karnataka, India; jashgowda20@gmail.com; 5Department of Pharmaceutical Chemistry, MM College of Pharmacy, Maharishi Markandeshwar (Deemed to Be University ), Mullana, Ambala 133203, Haryana, India; yousufniper@gmail.com; 6Department of Clinical Pharmacy, College of Pharmacy, King Khalid University (KKU), Abha 62529, Saudi Arabia; aishaa2804@gmail.com; 7Cancer Research Unit, King Khalid University, Abha 62529, Saudi Arabia; 8Department of Pharmacognosy, College of Pharmacy, King Khalid University (KKU), Abha 62529, Saudi Arabia; shad.nnp@gmail.com

**Keywords:** stomach cancer, pathophysiology, novel drug delivery systems, therapies

## Abstract

Cancer has long been regarded as one of the world’s most fatal diseases, claiming the lives of countless individuals each year. Stomach cancer is a prevalent cancer that has recently reached a high number of fatalities. It continues to be one of the most fatal cancer forms, requiring immediate attention due to its low overall survival rate. Early detection and appropriate therapy are, perhaps, of the most difficult challenges in the fight against stomach cancer. We focused on positive tactics for stomach cancer therapy in this paper, and we went over the most current advancements and progressions of nanotechnology-based systems in modern drug delivery and therapies in great detail. Recent therapeutic tactics used in nanotechnology-based delivery of drugs aim to improve cellular absorption, pharmacokinetics, and anticancer drug efficacy, allowing for more precise targeting of specific agents for effective stomach cancer treatment. The current review also provides information on ongoing research aimed at improving the curative effectiveness of existing anti-stomach cancer medicines. All these crucial matters discussed under one overarching title will be extremely useful to readers who are working on developing multi-functional nano-constructs for improved diagnosis and treatment of stomach cancer.

## 1. Introduction

Stomach cancer is the third main causative factor in morbidity and mortality worldwide. The forms of stomach cancer, which is also known as gastric cancer, are very heterogeneous from a morphologic standpoint. Age-related specificity indicates that gastric cancer starts at age 40 and peaks at age 75 [1]. Despite the decrease in incidents of gastric cancer in some areas there are more than about 1 million new cases and more than 78,4000 deaths annually reported globally [2]. The reduction seen in active gastric cancer and mortality is generally associated with a number of factors, such as less intake of salted, pickled, smoked, and chemically preserved nitrate-containing foods and increased consumption of fresh fruits and vegetables [3]. The main causative and triggering factor for stomach cancer is *Helicobacter pylori. H. pylori* is a known carcinogenic factor for non-cardia gastric cancer. Interestingly, the prevalence of gastric cancer variation appears with a broad geographical distribution, with the highest rates observed in Eastern Asia, Pacific Coast Southern America, and Eastern Europe and with lower rates in North America, the Northern region of Europe and Australia and some other individual countries also showing obvious clustering of stomach cancer cases [4,5]. Though there has been a decline in age-adjusted incidence rates in the past three to four decades, geographical invariancy remains at its peak. It has been observed that stomach cancer is most common among males. The reason for this might be associated with some associated risk factors, such as smoking, work stress, imbalanced diet, or some hormonal factors [6]. Certain conditions, such as improved economic status, maintained hygienic practices, and widespread use of good-quality improved foods, may play a certain role in dropping the rate of stomach cancer. Some recent molecular profiling and advancement in targeted therapy plays a vital and important role in the selection of drug therapy based on clinical patient studies. Figure 1 depicts the stages of stomach cancer.

## 2. Pathophysiology

Stomach cancer, which is the fifth most common type of cancer, has been known as a disease of many causes and the major known risk factor associated with it is *Helicobacter pylori* infection [7]. Another infection caused by Epstein–Barr virus (EBV) is also observed in stomach cancer cases which was almost 9.2% of the total death population in 2010 [8]. To understand the pathophysiology of stomach cancer it is necessary to reveal the structural organization of the human stomach. The main parts of the stomach are fundus, corpus, and pyloric antrum. The gastric mucosal represents three main types of glands, such as mucus-producing glands, oxyntic glands, and chief cells [9]. The main important steps in the recognition of gastric cancer are chronic atrophic gastritis and intestinal metaplasia (IM). Intestinal metaplasia is an initial signal of intestinal preneoplastic lesion which is generally identified by changes in the gastric mucosal region into false phenotype filled with goblet cells and intestinal mucins [10]. As discussed earlier, the main cause of gastric cancer is epigenetic modification in the tumor-suppressor genes which leads to uncontrolled cell proliferation apoptosis and rapid cell invasion. The additive factor *H. Pylori* infection results in gastritis and pyloris which may be considered a main and causative factor associated with carcinogenesis [11]. Though there is the role of extracellular microbes in stomach cancer, and food and diet-related habits also play an important role in the incidence of stomach cancer [12,13]. Some studies also suggest that sweet and salty balance in dietary consumption also plays a crucial role in the increased risk of gastric cancer [14]. The estimated mechanism of the role of dietary salt in the relative risk of gastric cancer is *H. Pylori* colonization and chronic inflammation [15,16]. Figure 2 depicts a schematic illustration of the role of Noxo1 and NOX1 in inflammation-associated gastric tumorigenesis.

## 3. Diagnosis and Therapies

### 3.1. Diagnosis

Patients in the later stages of stomach cancer usually present with symptoms such as nausea, vomiting, weight loss, abdominal pain, and peptic ulcer, whereas in the early stages most of them do not show any symptoms due to which early diagnosis is usually missed. Initially a double-contrast barium swallow, a cost-conscious, non-invasive, and a thorough study, is done [18,19,20]. The physician usually checks this radiographic study for preliminary information, such as the presence of a gastric lesion that may have benign or malignant features. If any vague results are reported or if signs of benign and malignant tumors are present, then further diagnostic evaluation is done. Esophago gastro duodenoscopy (EGD) is a specific and a very sensitive diagnostic test, mainly when performed in combination with endoscopic biopsy. Multiple biopsy specimens involving repeated sampling are taken from the suspicious site. Moreover, to accurately diagnose categories and as a decisive factor for neoadjuvant chemotherapy, endoscopic ultrasonography is necessary apart from EGD. Whereas, computed tomography is recommended for preoperative evaluation of tumors >T1, and for T3 and T4 tumors laparoscopy has become an efficient staging tool that also helps in detecting liver and peritoneal metastases [20,21]. The efficacy of endoscopic and radiographic examination for stomach cancer patients was evaluated by Matsumoto et al., who suggested that both these screening methods can help prevent tumor development [22]. Hamashima et al. reported a 30% reduction of death rate of stomach cancer patients that were screened by endoscopic examination when compared with a control group, within the three years before the diagnosis date of stomach cancer [23]. The improvements in the detection of stomach cancers have had a good effect on the clinical outcome of patients but the main issue is that stomach cancer is being diagnosed in the late stages when there is a high chance of metastasis and recurrence. To deal with this issue, a new technology of biomarkers for tumors has been adopted that helps in the early diagnosis of stomach cancer, the assessment of treatment efficacy, and in knowing the status of recurrence and metastasis [24]. Of the reported biomarkers for stomach cancer, i.e., carbohydrate antigen (CA)72-4, carbohydrate antigen (CA)12-5, BCA-225, alpha-fetoprotein, SLE, hCG, and pepsinogen I/II, carcinoembryonic antigen (CEA) and CA19-9 are the most common biomarkers in practical use. CEA is the marker most commonly used by clinicians [25]. It is also reported that increased concentrations of CEA are seen in the later stages of stomach cancer in a proportion of all stomach cancer patients; hence, it is not considered to be an efficient method of screening. CEA concentrations in peritoneal lavage fluid are said to precisely predict peritoneal recurrence after a curative surgery of stomach cancer [26]. An increased sensitivity has been reported by measuring immunohistochemical CEA according to the typical cytology method. Measuring CEA mRNA using RT-PCR is helpful to detect micrometastasis in the peritoneal cavity [27].

The other tumor biomarker, i.e., CA19-9, is a glycolipid antigen that was commonly used previously as a marker in gastrointestinal cancer [8]. In stomach cancer patients it might also be related to the depth and stage of tumor. Upon comparison with CEA, it was found that to estimate the tumor size, CA19-9 concentration in the serum is more diagnostically significant [28,29,30,31]. For stomach cancer patients, increased CA19-9 concentration can also indicate an early relapse after surgery and also metastasis in the peritoneal region [32,33], whereas a raised CA19-9 and (CA)72-4 serum concentration often point out an increase in the death rate among stomach cancer patients [34]. As per the reports of Song et al., an increase in the CA19-9 concentrations is mainly noticed in patients with stage III/IV group stomach cancer as compared to the I/II group [35]. Reports from the earlier studies state that the sensitivity for relapse of CA19-9 was 56%, with a specificity of 74% [36]. Usually, single tumor markers are not satisfactorily specific and sensitive so several markers are used in combination. In stomach cancer, CA19-9, CEA, carbohydrate antigen 15-3 (CA 15-3), and carbohydrate antigen 72-4 (CA 72-4) in serum have a significance in the early diagnosis and therapeutic monitoring of stomach cancer [37,38,39].

### 3.2. Chemotherapy

Stomach cancer treatment depends largely on the site of the tumor and how far it has spread. Apart from those factors, consideration has to be given to the patient‘s age, health status, and individual preference. Surgery is usually part of the typical treatment regimen because it provides the best chance for long-term survival. However, in cases where the patient cannot withstand it or if the cancer has widely spread it is not a preference. Other treatments such as chemotherapy and radiation therapy are usually a part of the treatment regimen, either in combination with or instead of surgery. Novel therapies such as targeted drugs and immunotherapy might also be of help in certain situations. Chemotherapy is the utilization of cytotoxic drugs to destroy the cancerous cells [40]. It is considered a primary treatment in metastasized cases of stomach cancer or if the cancer can’t be removed for some other reason. Chemotherapy helps in shrinking the tumor or slowing its growth and is usually given in cycles. It is used either before surgery, when it is called neoadjuvant treatment, or after surgery, when it is called adjuvant treatment. Neoadjuvant treatment minimizes the tumor size and, possibly, makes surgery easier, whereas adjuvant treatment can help in minimizing the recurrence of cancer. Usually post-surgery, stomach cancer patients’ chemotherapy is given in combination with radiation therapy thus helping treat tumors that were not removed totally by surgery [41,42,43,44,45,46,47].

Neoadjuvant chemotherapy is a novel method for treating advanced stomach cancer which aids in significant reduction of tumor stages, and even an improvement can be observed in the surgery success rate and patient survival time. Neoadjuvant chemotherapy can make resurgery possible for total tumor resection in laparotomy cases for unresectable stomach cancer [48]. Initially, it was reported by Wilke et al. that neoadjuvant chemotherapy can be used for the treatment of stomach cancer. Laparoscopic examination of 34 subjects with unresectable advanced stomach cancer led to their being given chemotherapy with adriamycin, etoposide, and cisplatin. Of those subjects, 33 had to undergo reoperation and then two cycles of post-surgical chemotherapy were performed with a reported remission rate of 70% [49]. In another study conducted by Crookes et al., 56 patients with advanced stomach cancer were given preoperative chemotherapy with 5-fluorouracil (5-FU) + calcium folinate + cisplatin (FLP), of whom 40 patients underwent radical resection, five reported complete remission, 12 cases were lowered to stage I, and 13 cases were lowered to stage II [50]. Whereas in another study of 24 advanced stomach cancer patients, the subjects were given a chemotherapy regimen with 5-FU + epirubicin + mitomycin (FAM) or MTX/5-FU. In 82% of patients, malignant ascites were gone, and radical resection was undertaken by 68% of the patients with a reported post-surgical median survival time of 14 months [51].

However, some reports have suggested that neoadjuvant chemotherapy can bring an improvement in the R0 resection rate and decrease the stages of tumor with no evident benefit in the long-term survival rate [52,53,54]. Pre-surgical chemotherapy using 5-fluorouracil/cisplatin enhanced the disease-free and overall survival of patients with advanced adenocarcinoma of the stomach and lower esophagus, as reported in a study conducted by Boige et al. [55].

When taking into account neoadjuvant chemotherapy for stomach cancer, regular review of treatment and screening indicators should be conducted, and resection would be considered as the optimal choice is there is a significant reduction in tumor size. Stomach cancer is relatively sensitive to chemotherapy drugs, and neoadjuvant chemotherapy and surgery are equally important for treatment and are greatly recommended in cases of limited metastatic stomach cancer [35,56,57].

Adjuvant chemotherapy is conducted to kill any areas of cancer that may have been left behind but are too small to see [25]. A meta-analysis study reported significant reduction in the death rate of stomach cancer patients on postoperative adjuvant chemotherapy with fluorouracil regimens in comparison with patients who undertook only surgery [58]. In patients treated with adjuvant chemotherapy, the overall survival increased from 49.6% to 55.3%, calculated for five years [59]. Yan et al. conducted a meta-analysis and systematic review to check the efficiency and safety of adjuvant intraperitoneal chemotherapy for patients with locally advanced resectable stomach cancer and reported that hyperthermic intraoperative intraperitoneal chemotherapy (HIIC), either with or without early postoperative intraperitoneal chemotherapy (EPIC) after the resection of advanced gastric primary cancer, is known to improve the overall survival rate but, sadly, higher risks of intra-abdominal abscess and neutropenia were also noted [60]. As per a few reports, it is known that adjuvant chemotherapy fetches a survival benefit in radically resected stomach cancer patients for stage ≥T2 [39,61,62].

### 3.3. Immunotherapy

In the last decade, gastric cancer (GC) is a leading cause of thousands of deaths and more than one million newly diagnosed cases worldwide [63]. There are a number of available therapeutic treatment options for targeted therapies and advanced treatments for better patient survival. Immunotherapy is a branch of immunology for better understanding new therapeutic methods which trigger the patient’s own immune system. This technique has been established recently and more attention has been given to advanced treatment of GC patients. Data published recently by D. Vrana et al. has established the importance of tumor immunology to therapy in gastric cancer as well as esophageal cancer. In this review article the authors compiled all relevant and possible treatment approaches for stomach cancer. The available reports suggest that the PD-1, PD-L1, PD-12 expression, and <MSI status for clinical prognosis and predictive role, together with potential clinical complications in stomach cancer treatment [64]. Whereas, D. Zeng et al. recently also published an article on the characterization of high tumor microenvironments in gastric cancer identification which have a potential role in immunotherapy for treating stomach cancer. There, about 1524 gastric cancer patients were taken in the studies and the conclusion was that the comprehensive landscape of the tumor microenvironment cells characterization of gastric cancer may help in the response of gastric tumors to immunotherapy and provide new advanced treatment of cancers [65]. Q. Zhano et al. have compiled the data of immunotherapy for gastric cancer treatment both as regards to dilemma and the prospective aspects of cancer treatment. In this review article they described the role of immunotherapy for the whole genomic sequence of personalized treatment to find a predictive biomarker and to help in the treatment of gastric patients in a safe way [66]. Coutzac et al. questioned this and made suggestions in their review article for future therapeutics. Immunotherapy offers real treatment for gastric cancer and they find a suitable approach and discuss all relevant aspects of treatment of gastric cancer. In this review article, Programmed cell death (PD-1) and its ligand (PD-L1) are seen as blocking by immune check point, with the result that they are seen as potential options for treating gastric cancer cells [67].

R. J. Kelly published a review article in which the author emphasised the role of PD-L1 up-regulation which occurs in approximately 40% of gastro and esophageal cancers. In this review article, Kelly compiled the roles of several approaches of immunotherapy and their successful treatments of esophageal and gastric cancers in which PD-L2 expression has been reported in 52% of esophageal adenocarcinomas. In this review, the author also compiled the different source data of immune microenvironments in diverse tumors which can explain responses or resistances to immunotherapy [68]. Z. Song et al. collected and compiled the data of all advanced treatments of gastric cancer and focused on combination therapies such as chemotherapy, molecular target therapy, and immunotherapeutic approaches and suggested that these therapies can help to attain five (5) years’ survival for earlier stages of gastric cancer, with a success rate of >95% [48]. J. N. Gerson et al. published a review article and identified the role of HER2-targeting in the treatment of gastric cancer as well as esophageal cancer. In this review article, the authors compiled the data from different research and review articles, both clinical and preclinical data, concluding that HER2 would be a good target for the immunotherapy approach for successful GC treatment [69].

F. M. Johnston et al. compiled and published a review article in which the main focus was the multidisciplinary approach in treating gastric adenocarcinomas because of recently advanced approaches and suitable treatments in gastric surgery but recurrence occurred more commonly when treating advanced gastric cancers [70]. Ramon Andrade De Mello et al. (2019) reported on immune checkpoint inhibitors (ICIs) for treating stomach malignancies. The ICIs such as nivolumab and pembrolizumab have been approved drugs for treating esophageal malignancies. Other ICIs, such as avelumab, durvalumab, etc., have been under clinical trial for the treatment of stomach cancers [71]. A. Pellino et al. also reported recent prospectives based on targeted therapies for the treatment of gastric cancer. In this review, the main emphases are on the global view of recent molecular diagnoses from the Cancer Genome Atlas and the Asian cancer research group and on the key promising developments in the field of immunotherapy and targeted therapies in metastatic gastric cancers [63].

### 3.4. Radiation Therapy

The treatment of gastric cancer cells by using radiotherapy is now a successful treatment. This approach is based on exposing high-energy X-rays directly to cancer cells which are present in the stomach or the cell lining of gastric cells. In this case a linear accelerator is used to generate the high-energy rays that are used to harm and demolish tumor cells present in the areas of treatment. This has a destructive effect on normal cells but they later regenerate normally. For better treatment, it is recommended that the high-energy treatment be interrupted which helps reduce damage to normal cells.

Recently, J. Tey et al. compiled a review article related to palliative radiotherapy for gastric cancer during 1995 to 2015 using data from medicine and a central search engine. Seven non-comparative observations were included in this article which concluded that most patients had good clinical benefits [72]. Audrey H. Choi et al. also published a review article which mentioned the importance of perioperative chemotherapy for the respective gastric cancers and concluded that the predictive molecular profiling of gastric patients needs to address and tailor therapies based on targeted genetic alterations [73]. Katherine E. Henson et al. published data in England during 2013–2014 which revealed socio-demographic variations on the use of chemotherapy and radiation therapy regarding patients. In this paper, 50,232 patients were identified for surveillance studies which found substantial variation for chemotherapy and radiotherapy in stage IV lung, esophageal, pancreatic, and stomach cancers [74]. Calin Cainap et al. published a review article about gastric cancer treatment by adjuvant chemotherapeutic and chemo radiation therapy and concluded that the current use of surgery with curative intent and adjuvant treatment based on efficacy and toxicity parameters is superior treatment for gastric patients [75]. G. Crehange et al. published a practical guideline regarding the use of radiotherapy in cancer treatment, especially esophagus, the gastric cardiac, and the stomach. The guideline clearly mentions the role of radiotherapy for better treatment or standardization protocol treatment of such cancers [76]. Xiaohui Pnag et al. reported the recent use of radiotherapy for gastric cancer and compiled data from different sources from a 10-year period. They also concluded that the effect of radiotherapy on five-year overall survival was also quite controversial. This also showed the use of radiotherapy for preoperative and postoperative radiotherapy [77].

## 4. Novel Drug Delivery Systems for Gastric Cancer Treatment

### 4.1. Nanotechnology Based Drug Delivery Systems

#### 4.1.1. Nanoparticles

Nanotechnology has phenomenally transformed the area of anticancer therapies, diagnosis, and drug delivery via NPs. NPs are commonly obtained by manipulating the particle structures of various polymers, inorganic materials, and organic materials in the 10–1000 nm size range [78]. These NPs are generally categorized into different types, such as polymeric NPs, metallic NPs, and metal-polymer nanocomposites, based on the materials used to fabricate them [79]. Some NPs aim to improve the efficacy of anticancer agents and other NPs themselves function as anticancer agents. However, all NPs actively target the tumor site with the aid of targeting moieties [80]. In addition, the morphological changes in the tumor microenvironment also allow NPs to passively target the tumor site. Nevertheless, these NPs can significantly overcome the current drawbacks associated with gastric cancer therapy [81].

#### 4.1.2. Polymeric Nanoparticles

The polymeric NPs are unique structures that can either encapsulate or embed the drug molecules to form nanocapsules or nanospheres. These polymeric NPs are most widely adopted as chemotherapeutic drug carriers due to their ability to improve cancer treatment by alleviating the drawbacks of drugs such as low solubility, poor permeability, short half-life, instability, and toxicity [82]. Nanocapsules are most preferably developed to protect the drugs from harsh physiological environments or to avoid the action of drugs on non-target sites. Conversely, nanospheres are preferred to control the delivery of drugs over a required period of time with necessary doses [83]. Many studies have investigated the polymeric NPs loaded chemotherapeutic drugs, such as 5-fluorouracil (5FU), docetaxel (DCT), paclitaxel (PCT), doxorubicin (DOX), etc., to treat gastric cancer. Recently, hybrid drug delivery (two or more chemotherapeutic agents) is being considered as a potential strategy despite delivering the individual drug to the tumor site [84]. However, precise delivery of multi drugs into the tumors is the greatest challenge even with NPs. This is due to the distinct physicochemical properties of anticancer drugs. Taking this into consideration, Hong and Feng developed polyethylene glycol and polylactide-coglycolide (PEG-PLGA) based NPs loaded with Irinotecan (SN-38) and 5FU (SN-38-5FU@NPs) via nano-precipitation method for the efficient treatment of gastric cancer [85].

For a list of nanoparticle type, anticancer drugs, polymers used, cell lines used, and application of different nanoparticles for the treatment of stomach cancer as novel drug delivery systems, see Table 1.

The developed SN-38-5FU@NPs was in the size range of 82–84 nm and polydispersity index of 0.147 ± 0.04 with more than 90% drug encapsulation efficiency. Further, SN-38-5FU@NPs exhibited more cytotoxic effects than free 5FU, SN-38, 5FU@NPs, and SN-38@NPs on two gastric cell lines, such as NCI-N87 and SGC-7901. The IC50 values for SN-38-5FU@NPs on NCI-N87 and SGC-7901 cell lines were 5.78 ± 0.86 µM and 7.16 ± 2.80 µM, respectively, whereas free drug and individual drug loaded NPs resulted in more than 10 µM IC50 values. The outcome of this study indicates that multi drug loaded polymeric NPs are the better choice for efficient gastric cancer therapy than individual drug loaded counterparts.

Abnormal activation of PI3K/AKT pathway regulates gastric cancer cells’ proliferation by inhibiting their apoptosis. Therefore, the pathway inhibitors, such as LY294002, are commonly adopted to treat gastric cancer by down-regulating MMP2, MMP9, and VEGF [95]. A few studies have shown increased gastric cancer treatment efficacy by combining the chemotherapeutic agents with PI3K/AKT inhibitors [95]. However, the short half-life of PI3K/AKT and poor solubility of DCT make them vulnerable candidates for efficient cancer therapy. To overcome this issue, Cai et al. have developed DCT and LY294002 loaded PLGA NPs [86]. The results from an in vitro cytotoxicity study on the MKN45 gastric cancer cell line depicted the improved antiproliferative effect of DCT-LY294002@NPs and plain DCT-LY294002 compared to plain DCT and LY294002. However, an in vivo study on xenograft nude mouse model exhibited the highest antitumor activity for DCT-LY294002@NPs compared to plain DCT-LY294002, DCT, and LY294002. This can be due to the enhanced accumulation of DCT-LY294002@NPs at the tumor site followed by controlled releasing ability.

Apart from developing drug encapsulated polymeric NPs alone, many studies have developed surface functionalized polymeric NPs to reduce the severe side effects of chemotherapeutic agents due to off-target drug delivery [96,97]. One such study involves the development of 5FU-PCT@PLGA NPs functionalized with anti-sLeA monoclonal antibody [87]. The fully functionalized NPs were in the size range of 137–330 nm. The results from the ex vivo study depicted the strong binding of functionalized NPs onto sLeA-expressing cancer cells by restricting the affinity towards normal healthy tissues. Thus, it can be concluded that functionalized NPs are potential candidates in targeted gastric cancer therapy.

Signal transducer and activator of transcription 3 (STAT3) is a master transcriptional factor that can regulate cancer cell proliferation. This protooncogene is commonly activated in many cancer conditions, including gastric cancer [98]. Studies have witnessed the STAT3-activated resistance of cancer cells towards numerous chemotherapeutic drugs, such as DOX, DCT, cisplatin, etc. To overcome this chemoresistance, researchers have made an attempt to utilize STAT3 inhibitors in conjugation with chemotherapeutic agents for efficient cancer therapy [99,100,101]. However, encapsulation of both STAT3 inhibitors and chemotherapeutic agents in single polymeric NPs could potentially bring down the off-target drug delivery and their side effects. On a similar note, Zheng et al. reported a nifuratel (NIF) and DOX loaded PLGA NPs for synergistic anticancer activity against two gastric cancer cell lines, such as SGC7901 and BGC-823 [102]. As depicted earlier, the concept of this study was to inhibit the STAT3 by NIF, which could preferentially enhance the activity of DOX against cancer cells without encountering resistance. The NIF and DOX loaded NPs (DNNPs) were in the size range of 148.18–229.39 nm. As expected, the NIF did not induce any cytotoxic effects on both the gastric cancer cell lines since it just inhibits STAT3. However, DNNPs exhibited maximum cytotoxicity against gastric cancer cell lines rather than plain DOX and DOX-NPs. Most interestingly, DOX-NPs was slightly more cytotoxic than plain DOX, which can be due to the increased uptake of NPs by cancer cells followed by controlled release of the drug over a longer period of time. Altogether, these results indicated that PLGA NPs are capable enough to be used as an efficient treatment against gastric cancers.

#### 4.1.3. Metallic Nanoparticles

Metallic NPs have gained significant attention as novel anticancer agents for gastric cancer therapy. These NPs are generally prepared by reducing and capping the metal precursors, such as silver nitrate, gold halides, zinc acetate dihydrate, copper nitrate, etc. [103]. Most commonly, the synthesis of metallic NPs includes hazardous chemicals and organic solvents as reducing agents, capping agents, and preparation medium. However, the green materials (natural), such as plant extracts, microbes, biodegradable waste materials, and water, have replaced the chemically synthesized metallic NPs leading to low-toxic, inexpensive, and eco-friendly NPs. Many studies have witnessed that the green synthesized metallic NPs possess lower cytotoxicity towards normal healthy cells compared to chemically synthesized counterparts [104]. Unlike polymeric NPs, metallic NPs can be utilized individually or combined with other drugs to exhibit synergistic anticancer activity against gastric cancer. Some of the metallic NPs, including zinc oxide, nickel oxide, cobalt oxide, silver, and gold NPs, have shown the most promising results against gastric cancer cells by facilitating better targeting, precise and controlled drug delivery, imaging, and gene silencing [105].

Recently, Tang and colleagues developed novel zinc oxide NPs using aqueous leaf extracts of *Morus nigra* as capping and reducing agents for gastric cancer therapy [88]. The prepared NPs were in the nano-size range and efficiently enhanced the anticancer activity against gastric cancer cell line (AGS) by deducing the mitochondrial membrane potential and increasing the intracellular ROS levels. As a result, these NPs successfully arrested the cell cycle leading to apoptosis. On a similar line, another study by Sun et al. has reported zinc oxide/neodymium nanocomposite using flower extract of *Cassia auriculata* L. as a reducing agent [106]. With an average size of 33.56 nm, the nanocomposite significantly arrested the cell growth and caused apoptosis in AGS cell line via inhibiting the PI3K/AKT/mTOR pathway. These results suggest that the zinc oxide/neodymium nanocomposite is a partially potential candidate for gastric cancer therapy. However, the preclinical studies would further confirm their complete potential.

Gold NPs are the most popular NPs due to their ability to both treat and diagnose cancer conditions. In addition, they are preferably less toxic and easy to synthesize in less time compared to other metallic NPs [107]. In a recent study, gold NPs were prepared using *Nigella sativa* (black cumin) seed extract and membrane vesicles of a *Curtobacterium proimmune* K3 (probiotic) [89]. The resulting NPs were in the size range of 30–50 nm as per TEM photomicrographs with an elliptical or polygonal shape. Further, the gold NPs depicted dose-dependent cytotoxicity against AGS cell line in the concentration range of 50–300 μg/mL and exhibited very slight toxicity towards normal healthy cell lines (RAW264.7 and HaCaT) in a similar concentration range. Results conclude that these NPs are effective against gastric cancer cells with good biocompatibility. An investigation on molecular mechanism underlying the cytotoxicity of these gold NPs against AGS cell line was conducted. The results revealed that up-regulation of apoptotic signaling and suppression of autophagy-related signaling pathways in AGS cell line was the main reason behind potential anticancer activity of gold NPs. Altogether, these results suggest that gold NPs can serve as novel anticancer candidates to treat gastric cancer.

Consequently, cobalt and nickel oxide NPs that were synthesized via chemical methods have shown dose-dependent cytotoxicity against AGS gastric cancer cell line. However, both cobalt and nickel oxide NPs exhibited slight toxicity towards normal fibroblast cells (L929) [90,108]. Nevertheless, silver NPs are one of the most widely used among many metallic NPs as an anticancer agent. In fact, the silver NPs synthesized via green route using numerous natural sources, such as *Scrophularia striata* [109], *Satureja Rechinger* [110], *Dysosma pleiantha* [111], *Artemisia ciniformis* [112], *Acacia nilotica* [113], and *Teucrium polium* [114], successfully hindered the proliferation of gastric cancer cells (AGS and MNK45) followed by inducing apoptosis leading to efficient gastric cancer therapy.

#### 4.1.4. Metal-Polymer Composite Nanoparticles

Both the polymeric NPs and metallic NPs have achieved adequate anticancer activity so far. The polymeric NPs were used to load and precisely deliver the anticancer drugs into the tumor site for efficient gastric cancer therapy. Whereas the metallic NPs, themselves kill the cancer cells due to their unique physicochemical properties [115]. In this regard, many researchers have speculated to combine metallic NPs (gold, silver, copper, magnetite NPs, etc.) and other anticancer agents, such as curcumin, resveratrol, DOX, DCT, PCT, etc., for synergistic activity against gastric cancer. To achieve this, the metallic NPs must be combined with polymers. This enables the conjugation of anticancer agents on metallic NPs. In one such research attempt, Yang et al. have reported the DOX, XMD8-92, and superparamagnetic iron oxide nanoparticles (SPIONs) loaded poly(ethylene glycol)-blocked-poly(L-leucine) (PEG-b-Leu) NPs (DXS@NPs) for gastric cancer therapy and imaging [91]. XMD8-92 is a novel chemo sensitizing agent, which can down-regulate P-gp in cancer cells to enhance the anticancer effect of DOX and SPIONs. In addition, biotin was also incorporated into DXS@NPs for tumor targeting and effective cellular uptake of NPs. The average size of the resulting NPs was found to be 105 nm. Furthermore, the DXS@NPs exhibited efficient tumor inhibition and highly declined systemic toxicity compared to free DOX and XS@NPs in gastric cancer-bearing mice (SGC-7901).

Apart from conjugating anticancer agents, the metal-polymer composites have also been actively involved in conjugating other metal NPs to result in metal–metal NPs. For instance, Wang and colleagues reported copper oxide NPs supported chitosan functionalized-amino magnetite NPs (Fe3O4-NH2@CS/CuO) for anti-gastric cancer effects [92]. The average particle size of the fully functionalized nanocomposite was 27.6 nm. The authors studied the anticancer activity of developed nanocomposite on three different gastric cancer cell lines, such as MKN45, AGS, and KATO III. The IC50 values against all three cell lines were 517 mg/mL, 525 mg/mL, and 544 mg/mL, indicating Fe_3_O_4_-NH_2_@CS/CuO nanocomposite is a versatile anti-gastric cancer agent. In a similar way, another nanocomposite was developed by Liu and team [116]. In this study, the authors developed Fe_3_O_4_/PEG2000/Cu nanocomposite by using green tea extract as a reducing and stabilizing agent. With an average particle size ranging from 20 to 40 nm, the Fe_3_O_4_/PEG2000/Cu nanocomposite effectively inhibited the proliferation of NCI-N87 and MKN45 gastric cancer cells, concluding that nanocomposites are remarkable candidates against gastric cancer conditions.

#### 4.1.5. Miscellaneous Nanoparticles

NPs are being extremely advantageous in terms of treating gastric cancer without inducing severe side effects. The chemotherapeutic agents are commonly associated with many drawbacks. Although NPs reduce their side effects to some extent by delivering them directly to the tumor site, the drug-related side effects cannot be completely eliminated [82]. Studies have shown that cancer cells possess unregulated levels of microRNAs-21 (miR-21), which can increase the proliferation rate of cancer cells by hindering their apoptosis. Therefore, an anti-miR oligonucleotide (AMO) is envisaged to be a potential candidate for cancer therapy by blocking certain signals to miR-21. In addition to this, many phytoconstituents have exhibited superior anticancer activity without inducing side effects [117]. Taking this virtue as a boon, some studies have made an effort to treat gastric cancer by using both AMO and phytoconstituents for their synergistic activity. However, delivery of both the therapeutic agents into the tumor site in an optimized concentration to achieve efficient gastric cancer therapy is a significant challenge. Therefore, in a research vocation, AMO and resveratrol (RSV) loaded mesoporous silica nanoparticles (MSNs) conjugated with hyaluronic acid (HA) was prepared to amplify gastric cancer therapy [93]. The purpose of functionalizing MSNs with HA is to target the over-expressed CD44 receptor on gastric cancer cells. The developed NPs released the loaded AMO-RSV in a sustained manner instead of initial burst release. Further, in vivo antitumor efficacy of developed NPs was determined using balb/c nude mice. A significant reduction in tumor size in the mice group treated with AMO-RSV@HA-MSNs was observed. Approximately, the AMO-RSV@HA-MSNs exhibited two-fold increased tumor regression effect compared with AMO-RSV@NPs. Therefore, this targeted delivery of dual therapeutic agents using functionalized inorganic NPs could be a promising strategy to enhance the therapeutic efficacy in gastric carcinoma.

Interestingly, a study by Asefi et al. reported the L-ascorbic acid (LAA) (Vitamin C) capped superparamagnetic iron oxide nanoparticles (SPIONs) for synergistic action on AGS (gastric cancer) cell line. The LAA-SPIONs induced increased apoptosis by altering the expression of p53 and Bcl-2 genes in AGS cell line compared to free LAA and SPI-ONs [118]. Similar to the previous two studies, another research team has made an attempt to develop chemotherapeutic agent and phytoconstituents loaded NPs for synergistic activity against gastric cancer through the reversal of chemoresistance [94]. Here, the authors have prepared cisplatin (CIS) and oleanolic acid (OA) loaded calcium carbonate (CC) NPs, further coated with cancer cell membrane (CM) to achieve targeted drug delivery. The developed CIS-OA@CM-CC NPs achieved pH-responsive (pH 5.5) sustained drug release for about 120 h. Further, the dual drug loaded NPs exhibited increased antitumor activity against MGC-803 tumor-bearing mice than individual drug-loaded NPs. These results clearly indicate that the combinatorial drug-loaded NPs outperform mono drug-loaded NPs in the treatment of gastric cancer.

### 4.2. Dendrimers

Nanomedicine is reflected as a latest and significant tool in current cancer management and treatment. There are several types of nanomedicine carriers, the most important being dendrimers. Dendrimers are used as nanocarriers which transport a wide variety of bioactive materials. Being derived from Greek words, “dendron” and “meros”, which means sensing tree and stem, dendrimers have an excellent property of carrying active guest molecules with controlled release in an efficient manner [119,120]. Dendrimers have been considered an important tool in biomedical applications in various drug-delivery therapies, such as sensing, and in vivo and in vitro imaging techniques. Dendrimers represent a unique class of macromolecules with nicely defined 3D architecture. Being different in shape when compared with conventional polymers, these are spherical with a controlled, and affording a high degree of, molecular uniformity. Size and shape are two important features to be kept in consideration for the design and development of biocompatible dendrimers [121]. Polyethylene glycol dendrimers are available after intravenous administration in the case of stomach cancer [122]. Dendritic polymers such as polyamino diamine (PAMAM), which is a specific type of dendrimer, have also gained some attraction for biomedical application [123]. The associated cytotoxicity of PAMAM limits its use as a diverse dendrimer [124]. PEGylated dendrimers which are loaded with celastrol which are obtained from the plant *Tripterygium wilfordii* are also widely used in treating gastric cancer.

### 4.3. Exosome

The exosome lies in the range of 30–150 nm (diameter), and its nanoscale vesicles are of endocytic origin which are secreted from all type of cells [125,126]. There are vital applications of exosomes for the treatment of GC. Recently, published work by Min Fu et al. presented the role and application of exosome in gastric cancer cell line treatment. In this review the article’s authors tried to find several applications and mechanisms on the biological role of exosomes and their potential as biomarkers for gastric cancer diagnosis as well as find a potential target for the therapy [127].

### 4.4. Liposomes

Liposomes were first reported in the year 1964 by a British hematologist, Dr. Alec D. Bangham, at the University of Cambridge. Liposomes are described as colloidal spherical vesicles composed of a phospholipid bilayer membrane that encapsulates a fraction of the surrounding aqueous medium. The amphiphilic nature of phospholipids promotes excellent cellular uptake due to the fact that they mimic natural cell membranes [128,129]. The size of liposomal vesicles ranges from 20–1000 nm based on the composition of lipid and the number of lipid bilayers. Further, the liposomes are mainly categorized into two types, unilamellar (containing single phospholipid bilayer) and multilamellar (containing more than one unilamellar separated by layers of water). The number of lamellae judges the drug-loading capacity of the vesicle. One of the major reasons that liposomes have gained significant interest in drug delivery is due to their ability to encapsulate both hydrophilic and lipophilic compounds in their aqueous center and lipid bilayer, respectively [130,131,132]. Liposomes possess excellent biocompatibility, biodegradability, non-toxicity, and non-immunogenicity. The vesicular encapsulation protects the drug from the physiological environment and also prevents the drug acting at the non-target site to circumvent toxicity, because of which they have been widely adopted to deliver numerous anticancer drugs for gastric cancer therapy [133].

Estrogen (ES) is a key component that has an impact on the growth and function of many systems. The function of estrogen is mainly due to the activation of estrogen receptors (ESRs), such as ESR-α and ESR-β. Recently, several studies have reported that gastric cancer expresses both ESR-α and ESR-β [134,135]. On this note, a research team from Jilin University has investigated the effect of ES-targeted PEGylated liposomes loaded with oxaliplatin (ES-SSL-OX) on gastric cancer [136]. The liposomes prepared via film hydration technique using soya phosphatidylcholine (SPC), cholesterol (CHO), and DSPE, and PEG2000 yielded an average particle size of 153.37 nm with 46.20% EE. The results from an in vitro cytotoxicity study on the SGC-7901 gastric cancer cell line showed maximum toxicity for ES-SSL-OX- compared to free OX. Later, the in vivo study on balb/c mice manifested superior suppression in tumor growth when treated with ES-SSL-OX without inducing significant side effects. Similarly, another study by the same research team has prepared mitoxantrone (MTO) loaded PEGylated liposomes for the targeted delivery of MTO to the gastric cancer cells via ESRs [137]. The results revealed that the PEGylated liposomes enhanced the in vivo circulation time of the drug leading to reduced drug metabolism and prolonged antitumor effect. Further, due to the estrogen targeting, the maximum amount of MTO had accumulated in the ESRs present on gastric tumor leading to significant increase in the antitumor effect of MTO followed by reducing its side effects due to off-target MTO delivery.

Nowadays, natural medicines have gained significant interest in gastric cancer therapy due to their low toxic profile compared with chemotherapeutic drugs [138]. Berberine (BBR) is an isoquinoline alkaloid derived from the herb *Coptis chinensis*. They have been widely used for the treatment of various gastrointestinal disorders, including cancer therapy. However, they possess certain side effects, such as hypotension, vasodilation, and cardiac suppression, when administered via IV route [139]. To overcome these side effects, Wang and colleagues have developed BBR-loaded PEGylated liposomes to improve the anticancer efficacy of BBR [140]. With an average particle size of 116.9 nm and Zeta potential of −31.4 mV, the PEGylated BBR liposomes endowed 45.8% of gastric tumor suppression, whereas non-PEGylated liposomes depicted 38.9% of tumor suppression. In addition to long-circulatory effect, the PEGylated liposomes exhibited sustained release of BBR for more than 48 h, due to which prolonged antitumor effect can be witnessed in the mice model.

Despite using individual drugs, some studies have been reported the usage of combination drugs such as natural drugs and chemotherapeutic agents for synergistic anticancer effect [141]. However, the delivery of an unstable natural drug and toxic chemotherapeutic drug to the gastric tumor is a challenging task. To overcome this difficulty, a study by Hong and his team have prepared two different liposomal formulations encapsulating ginsenoside (GC) and paclitaxel (PTX) for exerting a synergistic gastric cancer effect [142]. Interestingly, the authors have used GC as both a therapeutic agent and liposomal membrane stabilizer to improve the blood circulation of liposomal formulation. This combination therapy potentially suppressed gastric cancer tumor compared to most reported individual PTX formulations and the commercial product Abraxane^®^.

Although chemotherapeutic agents have captured the market of cancer therapy due to their potent tumor-killing ability, their toxicity towards normal cells has made researchers come up with targeted therapy [143]. Apatinib (AP) is a tyrosine kinase inhibitor that potentially inhibits the vascular endothelial growth factor receptor-2 (VEGFR2). Studies have shown promising results against gastric cancer when treated with AP with a better safety profile [144]. However, it exerts low oral bioavailability. However, the usage of high doses of AP to overcome the oral bioavailability issue leads to certain side effects, such as hypertension, hand–foot syndrome, and proteinuria. To improve therapeutic efficacy and reduce the side effects of AP against gastric cancer, an interesting study by Long and colleagues has developed pH-responsive liposomes loaded with AP [145]. Further to enhance the antitumor effect via synergistic action, cinobufagin (CS-1), a naturally occurring antitumor agent derived from toad, was impregnated into liposomes. Thereafter, to enhance the tumor-targeting ability and long-circulatory effect of LP@AC, a hybrid membrane derived from cancer cells (CCM) and red blood cells (RBC) was coated on LP@AC to obtain final nanocomplex, i.e., LP-R/C@AC. The entrapment efficiency for AP and CS-1 in the nanocomplex was found to be 94.2% and 99.9%, respectively. With the particle size and zeta potential of 108 nm and -7.5 mV, respectively, the developed LP-R/C@AC nanocomplex exhibited more than 90% cell viability in both smooth muscle cells line and human umbilical vein endothelial cell line at 60 µg/mL concentration indicating the good biocompatibility of the nanocomplex. The LP-R/C@AC exerted negligible AP and CS-1 release at pH 7.4, whereas gradual disassembly of the whole nanocomplex system leading to maximum drug release was observed at pH 5.2. Nevertheless, the LP-R/C@AC nanocomplex significantly suppressed the gastric tumor in balb/c mice compared to AP, CS-1, AC, and LP@AC. Based on this evidence, a long-circulatory tumor targeted liposomes loaded with natural drug and kinase inhibitors could be the potential candidates for effective gastric cancer therapy.

### 4.5. Polymeric Micelles

Polymeric micelles (PMs) are nano-sized colloidal particles whose size ranges from 10–1000 nm. PMs are generally formed via self-assembly of amphiphilic copolymers. Briefly, the amphiphilic copolymers exist as a single molecule in an aqueous solution below the critical micellar concentration (CMC). Further, a slight increase in CMC would lead to the self-assembly of these copolymers into micelles with a hydrophobic core and hydrophilic shell [146,147]. Some of the commonly used amphiphilic copolymers to form the micelles are polyethylene glycol-phosphatidylethanolamine (PEG-PE) (hydrophilic-hydrophobic), PEG-amino acids, PEG-carbonates, etc. These are also called di-block copolymers, which can, potentially, encapsulate the water-insoluble drugs in their hydrophobic core. However, the tri-block copolymers-based micelles that involve three polymers (hydrophilic-hydrophobic-hydrophilic) can potentially encapsulate both hydrophilic and hydrophobic drugs [148,149]. Recently, micelles have gained significant interest in various biomedical applications, such as drug solubilization, targeted drug delivery, stimuli-responsive drug delivery, improved bioavailability, etc. Most interestingly, the presence of a hydrophilic shell (PEG) hinders the uptake of micelles by the reticuloendothelial system (RES) making them available to treat the condition in the body for a longer period of time. Since the micelles are nano in size, they can extravasate through the altered endothelial cell junction followed by accumulation in the tumor microenvironment [150,151]. With all these benefits, a recent study by Liang and co-workers has developed docetaxel (DTX) loaded PEG-b-p(HPMAm-Bz) based Π electron-stabilized polymeric micelles to treat gastrointestinal cancer [152]. Further, the developed DTX@micelles exhibited an improved antitumor effect compared to plain DTX, indicating polymeric micelles are the suitable carrier system for gastric cancer therapy. In another investigation, a natural drug, i.e., emodin-loaded stearic acid-g-chitosan oligosaccharide (CSO-SA/EMO) micelles, were fabricated by Jiang and his team to alleviate the issues related to poor bioavailability of emodin in gastric cancer therapy [153]. Initially, the particle size was found to be 139.3 nm for CSO-SA micelles, and a gradual increase in the particle size up to 238.3 nm was observed after encapsulating EMO into CSO-SA micelles. By exhibiting maximum toxicity towards both MGC803 and BGC823 gastric cancer cell lines, the CSO-SA/EMO micelles showed significant tumor growth inhibition compared to free EMO.

In the quest of achieving combinatorial gastric cancer effect, Li et al. have developed paclitaxel (PTX) and tetrandrine (TDN) encapsulated micelles [154]. Further, the surface of micelles was modified with DSPE-PEG2000-cell penetrating peptide (CPP) and hyaluronic acid (HA) for long-circulatory and tumor targeting ability of micelles. The fully functionalized micelles were found to be having an average particle size of 90 nm, which is highly suitable for permeating gastric tumors. The in vitro study showed improved intra-cellular uptake of P/T@CPP/HA micelles. Furthermore, the results from in vivo study on the balb/c mice model exhibited maximum antitumor effect for the groups treated with P/T@CPP/HA micelles due to the synergistic activity of both PTX and TDN.

Most recently, a novel redox and pH-responsive oridonin (ORI) attached (ethylene glycol)-block-poly(L-lysine) based self-assembled micelles were developed by Xu et al. to overcome the poor aqueous solubility and low bioavailability of ORI, which further enhanced the gastric cancer therapy [155]. The developed ORI micelles were having an average size of 80 nm with the zeta potential of −12 mV. Also, the maximum ORI that was loaded into the micelle was 18.7%. The in vitro drug-release study depicted maximum release of ORI at high GSH concentration and low pH level. Due to the presence of PEG, a significant increase in the circulation time of ORI@micelles in the body was observed. Further, by the end of 12 h, maximum accumulation of ORI@micelles at the tumor site was witnessed. Finally, the highest gastric tumor suppression was observed for the balb/c mice treated with ORI@micelles over free ORI and redox-insensitive ORI@micelles. Altogether these results indicate that dual-responsive ORI@micelles could be the potential system for treating advanced gastric cancer conditions. A list of liposomes/micelles, anticancer drugs, polymers/lipids used, cell lines used, and application of liposomes/micelles for the treatment of stomach cancer as novel drug-delivery systems is shown in Table 2.

## 5. Other Delivery Systems

### 5.1. Hydrogels

Hyaluronic Acid (HA) hydrogels were first used in humans mainly for pain in osteoarthritis as a viscoelastic fluid and as sheet formulations for preventing surgical adhesions. Currently, these HA hydrogels, because of their desirable properties, such as adaptive chemistry, biodegradability, biocompatibility, viscoelasticity, and chondrogenic potential, have found applications in tissue engineering and regenerative medicine [159], separation of biomolecules or cells, and regulation of biological adhesions as barrier materials leading to their use in anticancer therapy [160]. Chemical functionalization or crosslinking reactions are required to improve stability and to ensure the accuracy of the shape of the hydrogel. In vivo, the resilience of HA hydrogels depends on their rate of degradation by hyaluronidases and reactive oxygen and nitrogen species, which can limit their effectiveness [161]. Ravichandran et al. developed pH-sensitive, biocompatible, bio-degradable hydrogels for site-specific drug delivery in the gastric environment of the stomach. They prepared poly[NVP-AA]-PEG inter polymeric hydrogels and entrapped in it anticancer drug, 5-florouracil and carried out the in vitro release studies of the entrapped drug in SGF. Their findings could be applied for localized drug delivery in the acidic environment of the stomach [162]. M. Zhou et al. have demonstrated chemo-photothermal therapy for treating gastric cancers. Initially, they found that single-walled carbon nano-tube (SWNT) hydrogel is nontoxic on gastric cancer cells (BGC-823 cell line) but with NIR radiation leads to cell death through hyperthermia pro-apoptosis mechanism. They developed doxorubicin (DOX) loaded SWNT hydrogel and used with NIR radiation on mice xenograft gastric tumor models. The result of this hydrogel was the improved efficacy of doxorubicin without any organ toxicity when compared to free DOX. This developed material has potential and could be used for the treatment of gastric cancers [163].

S. Emoto et al. in their study encapsulated cisplatin in HA polymers and evaluated it for peritoneal dissemination of gastric cancer through intraperitoneal administration. This hydrogel was found to be an effective biomaterial as it was retained for a long time in the peritoneal cavity which could be attributed to the sustained release of cisplatin from the HA hydrogel and it also showed enhanced antitumor effects [164]. In a recent study, Li et al. prepared hydrogels by combining dialdehyde-modified HA (AHA) with cystamine dihydrochloride at low pH to treat gastric and mammary tumors. They demonstrated that the hydrogel morphology, swelling, and kinetics of gelation could be controlled by varying Cys-to-AHA ratio. They found that the mechanical properties of this hydrogel were improved when cystamine content was increased. They have concluded that this hydrogel could provide a promising future for various biomedical applications in drug delivery, bioprinting, smart robots, and tissue regeneration [165].

### 5.2. Microbubbles

Drug-loaded microbubbles can be used in anticancer therapy as these microbubbles can be destroyed at the focus position by ultrasound irradiation to facilitate the release of a drug directly at the focus position. This drug-delivery method increases the local drug concentration at the focused sites and, hence, decreases the exposure dose to non-focal regions, thus reducing cytotoxicity and side effects. Various coating materials, including lipids, surface active agents, proteins, and polymers, have been used to attach drugs to these microbubbles [166]. They can be adhered to the surface of the microbubbles, wrapped inside them, or combined with the membrane by noncovalent bonds [167]. Hydrophobic drugs, such as doxorubicin (DOX), paclitaxel (PTX), and docetaxel, have been successfully incorporated into microbubble shells [168]. After ultrasound irradiation, these microbubbles burst in the tumor tissue and the drugs carried by them are released directly into it.

B. Lai et al. evaluated the effect of docetaxel-loaded lipid microbubble (DLLD) in combination with ultrasound-triggered microbubble destruction (UTMD) on the growth of a gastric cancer cell line [169]. It was observed that the combined treatment effectively inhibits the growth of a gastric cell line, through cell-cycle arrest, promotion of apoptosis, etc. The authors explained in their study that the inhibition of cell growth by this DLLD system is mediated by the activation of p53 which is a well-characterized molecule that mediates cell-cycle arrest and cell apoptosis. This hypothesis of cell-growth inhibition by p53 activation in treating colorectal cancer with docletaxel is also documented [170]. The significant role of p53 as an important tumor-suppressor gene in humans is demonstrated very well in one recent clinical study. The results of the study revealed that the combination of SGT53 with docetaxel showed enhanced antitumor activity compared to docetaxel alone [171]. In a very recent study, Sun Li et al. have discussed the combination of sonodynamic (SDT) and antibody therapy for the treatment of HER2-positive gastric cancer. For their study, they prepared microbubble from pyropheophorbide-lipid, a sonosensitizer, conjugated it with trastuzumab (TP), and final encapsulation with SF6 gas. This developed microbubble simultaneously releases sonosensitizers and therapeutic antibodies at the tumor tissue upon irradiation with ultrasound. In their study, they observed that ultrasound-mediated TP MBs increase the uptake of sonosensitizers and also generate singlet oxygen which has a killing effect on cells and inhibits tumor growth with an inhibition rate of up to 79.3%. They concluded that this combined therapy has synergistic anticancer activity both in vivo and in vitro and holds great potential in improving the therapeutic efficiency on HER2-positive gastric cancer [172].

### 5.3. Microparticles

Microparticles have dimensions of 0.1–100 µm in diameter and are generally formulated for sustained release of a drug by microencapsulation techniques. PLGA based microparticles are widely applied as drug carriers as they are biodegradable and biocompatible. M. Boisdron-Celle and co-workers prepared PLAGA microparticles of 5-fluorouracil (5-FU) by an emulsion–extraction process as a drug-delivery system in the management of cancers. During their study, they observed that release of 5-fluorouracil is influenced by the morphology of the particles, 5-fluorouracil crystal size, and the PLAGA concentration [173]. J. L. Au and colleagues developed drug-loaded, tumor-penetrating microparticles (TPM) to overcome the problems associated with intraperitoneal chemotherapy in the treatment of peritoneal malignancies, including gastric cancers. In order to achieve the desired properties, such as tumor priming, selectivity, enhanced particle penetration, greater retention, and immediate and sustained antitumor activity, they used PLGA or poly(lactic-co-glycolic acid) polymers to design TPM. The particle size of TPM was fixed in the range of 4–6 μm to promote intra-cavity distribution. Preclinical studies revealed that paclitaxel-loaded TPM is more effective, less toxic, requires less frequent dosing, and has broad-spectrum activity against several IP metastatic tumors with different characteristics when compared to the intravenous paclitaxel/Cremophor micellar solution [174]. Overall, these studies have suggested that a suitable drug-delivery system in the form of microparticles conjugated to chemotherapeutic drugs can be applied for the improvement of the efficiency of anticancer drugs in the treatment of gastric cancer.

### 5.4. Oral Delivery

Oral drug delivery is the most convenient and less invasive route. However, it is very challenging because of the gastric acidic condition and the difficulty of localizing it in a selected region of the GI tract. In order to overcome these issues, many types of oral controlled drug-delivery systems having extended gastric retention times have been reported such as floating drug-delivery systems (FDDS), mucoadhesive systems, and other delayed gastric emptying devices. FDDS offers great advantage as it remains floating in the gastric fluid and the drug is released slowly from the system at a desired rate.

Yu Huang and co-workers developed 5-FU hollow microspheres by mixing polymer blends of PVP–EC to improve its oral bioavailability. This floating drug-delivery system showed a high drug-loading amount of approx. 28.4%, excellent floating, enhanced oral bioavailability, and sustained release characteristics in simulated gastric and intestinal fluids. In order to attain uniform particle size distribution, 1.5% of Span 80, an emulsifier, was selected which prevented coalescence. They also investigated biodistribution of the drug in tumor-bearing nude mice after oral administration of 5-FU hollow microspheres. The results revealed that the animals administered with 5-FU hollow microspheres had much higher drug content in tumor, plasma, and stomach at 1 and 8 h in comparison with those administered with 5-FU solid microspheres and its powder. Their findings could serve as a promising sustained and controlled drug-delivery system for an oral chemotherapy agent such as 5-FU [175].

M. Bar-Zeev et al. investigated the versatility of β-casein-based delivery system using different synergistic drug to treat MDR gastric cancer cells. The authors noted that chemotherapeutic drug SN-38 showed high binding affinity to casein-based nano vehicles and confirmed that β-casein solubilized these hydrophobic drugs. Their findings could serve as an efficient platform for oral delivery, local target-activated release of synergistic hydrophobic drug combinations to treat gastric cancer and to overcome cancer chemo resistance [176].

Bhardwaj and co-workers developed oral floating hollow microspheres bearing 5-Fu using Eudragit S100 as polymer by modified solvent evaporation technique and evaluated the hollow microspheres for its micromeritic properties and performed in vitro drug and in vivo studies. In their study, they found that this novel drug-delivery system showed good floating ability up to 18 h and the cumulative drug release between 81 and 96%. They concluded from in vitro drug release studies that release of the drug can be controlled by changing the ratio of polymer and solvent. It is also confirmed from in vivo, X-ray radiographic studies that the prepared dosage forms are retained in GIT for a long period of time. Their findings could serve as an efficient oral delivery of 5-FU to treat stomach cancers [177]. Table 3 lists anticancer drugs, polymers, and cell lines used in drug-delivery systems for the treatment of stomach cancer, with their advantages.

## 6. Conclusions and Future Perspectives

Stomach cancer is indeed a malignancy with a high fatality rate among the many cancer types. The primary goal of any effective cancer therapy is to destroy malignant cells while minimizing harm to healthy cells. Despite tremendous improvements and progress in cancer operations and adjuvant medications, stomach cancer allied mortality remains high, indicating that there is still scope for research in advancing therapy. We primarily covered several promising therapeutic strategies and efforts to generate new, cost-effective drugs for treating and managing stomach cancer in this stomach cancer review. The current review suggests that new-fangled drug-delivery approaches and competent systems, such as the incorporation of nanotechnology-based approaches into and alongside radiotherapy, immunotherapy, gene therapy, and other equivalent therapeutic strategies, be used to effectively eradicate stomach cancer. A combination of localized therapy and tumor targeting could well be the best technique for successfully managing stomach cancer. As a whole, these innovative nanocarrier-based therapeutic systems containing anticancer medications may provide the most convenient option of drug localization and delivery to targeted sites. These delivery techniques effectively halt the progression of cancer cells while also aiding in their apoptosis. In conclusion, the nanotechnology platform has immense potential, and recent in vitro and/or in vivo findings show that it is an emerging branch of science with numerous applications in drug delivery and therapies. However, more in vivo studies are required to confirm and evaluate the safety, toxicity, and targeted tumor-drug delivery of these reported nanocarriers before they can be used in clinical trials in the near future. In summary, nanocarrier-based novel-cum-advanced drug-delivery systems are, indeed, effective, precise, and efficient in the treatment of stomach cancer, and they hold substantial promise.

## Figures and Tables

**Figure 1 pharmaceutics-14-01576-f001:**
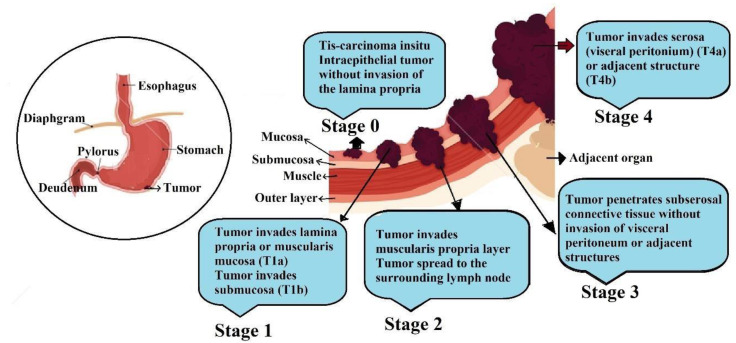
Different stages (0 to 4) of gastric cancer from a normal cell to the proliferation of cancer cells and spreading a cancerous cell into the bloodstream from the inner part to the outer part.

**Figure 2 pharmaceutics-14-01576-f002:**
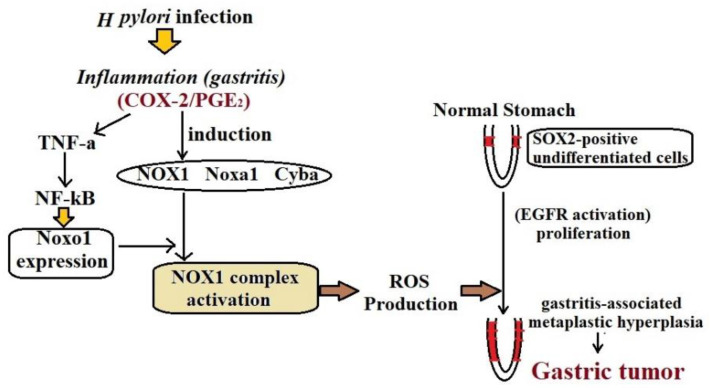
Role of Noxo1 and NOX1 in the gastric tumorigenesis in the *H. pylori*-infected stomach [17].

**Table 1 pharmaceutics-14-01576-t001:** List of nanoparticle type, anticancer drugs, polymers used, cell lines used, and application of different nanoparticles for the treatment of stomach cancer as novel drug delivery systems.

Type of Nanoparticles	Drug	Polymers/ Capping/ Reducing Agents	Cell Line/ Animal Model	Application	Ref.
Polymeric nanoparticles	Irinotecan and 5-fluorouracil	polyethylene glycol and polylactide- coglycolide	NCI-N87 and SGC- 7901 (human gastric cancer cell lines)	To establish synergistic chemotherapy followed by reducing the chemotherapeutic agent related side effects	[85]
Polymeric nanoparticles	Docetaxel and LY294002	Polylactic- coglycolic acid	MKN45 (human gastric cancer cell line)/ tumor-bearing Balb/c nude mice	To enhance the anticancer efficacy of docetaxel by inhibiting the PI3K/AKT pathway using LY294002	[86]
Polymeric nanoparticles	5-Fluorouracil and paclitaxel	Polylactic- coglycolic acid	NCI-N-87 and AGS (human gastric cancer cell line)	To achieve tumor targeted delivery of chemotherapeutic agents using anti-sLeA monoclonal antibody as a targeting moiety for improved gastric cancer efficacy	[87]
Metallic nanoparticles	Zinc oxide nanoparticles	Aqueous leaf extract of *Morus nigra*	AGS (human gastric cancer cell line)	To achieve anti-gastric cancer effects	[88]
Metallic nanoparticles	Gold nanoparticles	*Nigella sativa* (black cumin) seed extract and membrane vesicles of a *Curtobacterium proimmune* K3 (probiotic)	AGS (human gastric cancer cell line), RAW264.7 and HaCaT (normal healthy cell line)	To improve the gastric cancer therapy and to overcome the biocompatibility issues associated with chemically synthesized gold nanoparticles	[89]
Metallic nanoparticles	Nickel oxide nanoparticles	Glutamic acid and thiosemicarbazide	AGS (human gastric cancer cell line)	A novel therapeutic modality for gastric cancer	[90]
Metal- polymer composite nanoparticles	Doxorubicin, XMD8-92 (chemosensitizing agent), and superparamagnetic iron oxide nanoparticles	Poly(ethylene glycol)-blocked- poly(L-leucine)	Gastric cancer-bearing balb/c nude mice (SGC-7901)	To achieve synergistic anti-gastric cancer activity by down- regulating P-gp in gastric cancer cells	[91]
Metal-polymer composite nanoparticles	Copper oxide nanoparticles and magnetite nanoparticles	Chitosan	MKN45, AGS, and KATO III (human gastric cancer cell line)	Synergistically suppress the gastric tumors via two metallic nanoparticles	[92]
Mesoporous silica nanoparticles	Resveratrol and anti-miR oligonucleotide	Cetyltrimethylammonium bromide and hyaluronic acid	Gastric cancer induced male balb/c nude mice (BGC823)	To enhance the anticancer efficacy of resveratrol by inhibiting the microRNAs-21, which is responsible for cancer cell proliferation	[93]
Calcium carbonate nanoparticles	Cisplatin and oleanolic acid	Cancer cell membrane and calcium carbonate	Gastric cancer bearing male balb/c nude mice (MGC-803)	To overcome chemoresistance to cisplatin in gastric cancer	[94]

**Table 2 pharmaceutics-14-01576-t002:** List of liposomes/micelles, anticancer drugs, polymers/lipids used, cell lines used, and application of liposomes/micelles for the treatment of stomach cancer as novel drug-delivery systems.

Drug	Polymers/ Lipids	Cell Line/ Animal Model	Application	Ref.
		**Liposomes**		
TSPAN1 siRNA	1, 2-dioleoyl-3- trimethylammonium-propane, (DOTAP), avanti polar lipids, DSPE-PEG-Mal and cholesterol	Th17 cells/gastric tumor bearing hybrid mice	To decrease in CD4+ T cells polarization to Th17 cells followed by inhibition of gastric tumor formation	[152]
ubiquitin- specific proteases-22 (USP22) siRNA	DOTAP, DSPE-mPEG and DSPE- PEG-Mal, and cholesterol	MKN-45 (human gastric cancer cell line)/gastric cancer induced male balb/c nude mice	To improve the therapeutic efficacy of USP22 siRNA against gastric tumor with the help of CD44 antibodies	[153]
Special AT-rich sequence binding protein 1 (SATB1) siRNA	DOTAP, DSPE-mPEG, and DSPE-PEG-Mal, and cholesterol	MKN-45 and NCI-N87 (human gastric cancer cell line)	To enhance the therapeutic efficacy of SATB1siRNA against gastric tumor with the help of CD44 antibodies	[154]
Mitoxantrone	Phosphatidylcholine, DSPE- mPEG2000, and cholesterol	Tumor induced female balb/c nude mice	To reduce the side effects of mitoxantrone followed by enhancement of gastric cancer therapy via targeted delivery	[137]
Berberine	Hydrogenated soy phosphatidylcholine, 2000-(polyethylene glycol) distearoyl phosphatidyl ethanolamine (PEG2000-DSPE), and cholesterol	SGC-7901 (human gastric cancer cell line)/gastric cancer bearing balb/c nude mice (SGC-7901)	To reduce the side effects of berberine followed by enhancement of gastric cancer therapy via targeted delivery	[140]
		**Micelles**		
Paclitaxel	NH2-PEG-OH and 3,3′- Dithiodipropionic acid	SGC-7901 (human gastric cancer cell line)/gastric cancer bearing female balb/c nude mice (SGC-7901)	To achieve gastric tumor targeted controlled delivery of paclitaxel for effective gastric cancer therapy	[155]
CKR12 peptide (LL- 37 peptide fragment analog)	Polylactic co-glycolic acid and 3-(2-pyridyldithio) propionyl hydrazide	-	To improve the permeability of CKR12 peptide leading to the improvement of anti-gastric cancer effect	[156]
Doxorubicin	Heparosan-cystamine-vitamin E succinate	MGC80-3 (human gastric cancer cell line)	To enhance the anti-gastric cancer effect of doxorubicin with the help of redox- responsive drug delivery	[157]
Paclitaxel	Vitamin B12, sericin, synthetic poly(γbenzyl-L-glutamate)	BGC-823 (human gastric cancer cell line)	To improve the gastric cancer therapy by achieving targeted delivery of paclitaxel	[158]

**Table 3 pharmaceutics-14-01576-t003:** List of anticancer drugs, polymers, and cell lines used in drug-delivery systems for the treatment of stomach cancer with their advantages.

Drug + System	Polymer Used	Cell Line	Application	References
**Hydrogels**				
Doxorubicin-loaded single wall nanotube thermo-sensitive hydrogel for gastric cancer chemo-photothermal therapy	NA	BGC-823 cell line	Efficacy and lesser toxicity	[163]
Intraperitoneal administration of cisplatin via an in-situ cross-linkable hyaluronic acid-based hydrogel for peritoneal dissemination of gastric cancer	NA	MKN45P, a human gastric cancer cell line	Sustained drug delivery	[164]
**Microbubble**				
Docetaxel-loaded lipid microbubble (DLLD) in combination with ultrasound-triggered microbubble destruction (UTMD) on the growth of a gastric cancer cell line	JC-1	BGC-823	More efficient in inhibiting cell proliferation and inducing cell apoptosis in the gastric cancer cell line	[169]
Ultrasound Microbubbles Mediated Sonosensitizer and Antibody Co-delivery on HER2- Positive Gastric Cancer	NA	HER2-positive gastric cancer NCI-N87 cells	Significant tumor lethal effect in vitro and distinctly inhibited tumor growth in vivo	[172]
**Microparticles**				
5-fluorouracil-loaded microparticles as biodegradable anticancer drug carriers	biodegradable poly ((±)-lactide-co-glycolide) (PLAGA)	NA	Sustained drug delivery	[173]
Drug-loaded microparticles for treatment of peritoneal cancer	PLGA or poly (lacticco-glycolic acid) copolymer	NA	Less toxic and more effective against several IP metastatic tumors	[174]
**Oral drug delivery**				
5-fluorouracil-loaded floating gastroretentive hollow microsphere	polyvinyl pyrrolidone (PVP) and ethyl cellulose (EC) as drug controlled- release polymer blends.	MCF-7 breast cancer cells to induce tumor in mice	5-FU hollow microspheres exhibited excellent floating and sustained release characteristics.	[175]
Re-assembled casein micelles for oral delivery of chemotherapeutic combinations to overcome multidrug resistance in gastric cancer	NA	Human MDR gastric carcinoma cell line	Casein-based oral delivery systems provide a robust natural platform enabling a spectrum of development possibilities for gastric-activated release of synergistic drug combinations Developed oral drug delivery system showed good floating ability and it retained in GIT for a prolonged period of time.	[176]
Site Specific Hollow Floating Microspheres Bearing 5-Fu	Eudragit S-100	NA	system showed good floating ability and it retained in GIT for a prolonged period of time.	[177]

## Data Availability

Will be provided upon request.

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
