# Peer review of "Novel Drug Delivery Systems as an Emerging Platform for Stomach Cancer Therapy"

_pharmaceutics, 2022, doi:10.3390/pharmaceutics14081576_

Round 1
Reviewer 1 Report
Overall, the review “Novel drug delivery systems as an emerging platform for stomach cancer therapy” entitled provides a comprehensive summary of the currently applied therapies in gastric cancer and emerging approaches involving nanotechnology based drug delivery systems. This review could be of scientific interest to researchers working in the field of drug discovery and gastric cancer.
The manuscript is well structured; the sections discussing the different nanotechnology-based drug delivery systems provide detailed information on current progress in the development of these systems. An adequate number of references are included in the review.
There are some minor points that should be addressed to improve the manuscript. The figures seem to be of poor quality (resolution). Some text is not readable in Figures 3 and 4. Panel 4B has panels a, b, and c which is confusing.
Reviewer 2 Report
The review manuscript entitled "Novel drug delivery systems as an emerging platform for stomach cancer therapy” involved the bibliographical analyses of several therapeutic strategies based on drug delivery systems.
The introduction is complete and according to the developed topic of the manuscript, and it has updated bibliographical references to support the research.
Also, the manuscript is interesting, clear, organize, and focused on the topic that is of growing interest due to the potential uses of DDS to treat a diversity type of cancer.
Furthermore, the information they described is supported with clear and logical images/figures/tables that summarize all the obtained information and data.
I strongly suggest including the chronological developments of the reported strategies in order to understand the advancement of them through the years. This timeline could help the readers to focus on the most important and current developments.
Furthermore, I encourage the authors to check some mistakes (in yellow, pdf file attached) such as:
- Please I would like to encourage the authors to check the whole manuscript due to several paragraphs with different size/style (letters). Some of them are highlighted in the manuscript review file.
-Please check the use of 5-FU abbreviation because sometimes the authors wrote the full name and sometimes only the abbreviation. Remember the first time the authors wrote this compound should be 5-fluorouracil (5-FU) and then they can use only the 5-FU abbreviation.
- Please remember to use italics for in vivo and in vitro.
- There are several blank spaces throughout the manuscript and some words appear without the appropriate spaces between them.
- Please homologate the size of the graphics throughout the manuscript and the style of them. Moreover, it is important to improve the quality of these figures because the letters are all blurry (example: figure1 1, 3, and 4)
- Please check all the references because they should be according to the Pharmaceutics guidelines (https://www.mdpi.com/journal/pharmaceutics/instructions#preparation)
Example:
[1] E.C. Smyth, M. Nilsson, H.I. Grabsch, N.C. van Grieken, F. Lordick, Gastric cancer, The Lancet, 396 (2020) 635-648.
It should be:
[1] Smyth, E.C.; Nilsson, M.; Grabsch, H.I.; van Grieken, N.C.; Lordick, F.; Gastric cancer, Lancet 2020, 396, 635-648.
(As an example: the abbreviation of The Lancet is Lancet,)
Additionally, it is important to note that the conclusion part would be useful for any kind of cancer. Considering the topic of this review, I suggest improving this part specifying the importance of the reported DDS for stomach cancer.
I would like to invite the authors to add the abbreviation list of words at the end of this manuscript.
I recommend the acceptance of this manuscript after the authors performed the suggested corrections/additions.

Round 2
Reviewer 2 Report
The authors performed all the suggested corrections and modifications.